# Three Methods for Application of Data from a Volumetric Method to the Kissinger Equation to Obtain Activation Energy

**DOI:** 10.3390/mi13111809

**Published:** 2022-10-23

**Authors:** Myoung Youp Song, Young Jun Kwak

**Affiliations:** Division of Advanced Materials Engineering, Hydrogen & Fuel Cell Research Center, Engineering Research Institute, Jeonbuk National University, 567 Baekje-daero, Deokjin-gu, Jeonju 54896, Korea

**Keywords:** hydrogen storage, volumetric method, Sieverts-type volumetric apparatus, Kissinger equation, activation energy

## Abstract

Thermal analysis methods have been used in many reports to determine the activation energy for hydride decomposition (dehydrogenation). In our preceding work, we showed that the dehydrogenation rate of Mg-5Ni samples obeyed the first-order law, and the Kissinger equation could thus be used to determine the activation energy. In the present work, we obtained the activation energy for dehydrogenation by applying data from a volumetric method to the Kissinger equation. The quantity of hydrogen released from hydrogenated Mg-5Ni samples and the temperature of the reactor were measured as a function of time at different heating rates (*Φ*) in a Sieverts-type volumetric apparatus. The values of d*H*_d_/d*t*, the dehydrogenation rate, were calculated as time elapsed and the temperature (*T*_m_) with the highest d*H*_d_/d*t* was obtained. The values of d*H*_d_/d*T*, the rate of increase in released hydrogen quantity (*H*_d_) to temperature (*T*) increase, were calculated according to time, and the temperature (*T*_m_) with the highest d*H*_d_/d*T* was also obtained. In addition, the values of d*T*/d*t*, the rate of increase in temperature to time (*t*) increase, were calculated according to time, and the temperature (*T*_m_) with the highest d*H*_d_/d*t* was obtained. *Φ* and *T*_m_ were then applied to the Kissinger equation to determine the activation energy for dehydrogenation of Mg-5Ni samples.

## 1. Introduction

Hydrogen has drawn attention as a next-generation energy carrier. To use hydrogen as energy carrier, hydrogen-storage materials should be developed. Metal hydrides such as hydrides of Mg-based alloys [1,2,3,4] are one of the candidates for hydrogen-storage materials. Pure Mg absorbs and releases hydrogen very slowly. We added a small amount of Ni (5 wt%) to make the samples have dehydrogenation rates high enough to measure the dehydrogenation rate easily in this work.

Thermal analysis methods—such as thermogravimetric analysis (TGA), differential scanning calorimetry (DSC) analysis, differential thermal analysis (DTA), and thermal desorption spectroscopy (TDS) analysis—have been used in many reports to determine the activation energy for hydride decomposition (dehydrogenation) [5,6,7,8,9,10,11,12].

For the thermal analyses, the samples should be transferred to thermal analysis equipment after hydrogenation. To prevent the samples from being oxidized, equipment to isolate the hydrogenated samples is required, or the volumetric hydrogenation equipment and the thermal analysis equipment should be kept in an inert atmosphere simultaneously. Analysis using thermal analysis equipment is not always easy to be performed because evolved hydrogen is considered to contaminate parts of sensors for thermal analysis equipment. In the present work, we obtained the activation energy for the dehydrogenation of hydrogenated Mg-5Ni samples by applying data from a volu metric method to the Kissinger equation. This method has an advantage: data can be obtained in the same apparatus where samples were activated and hydrogenated.

In order that the Kissinger equation can be used to obtain the activation energy, the reaction rate equation must obey a first-order law [13]. However, the previous works obtained the activation energy were performed without showing obedience to the first-order law by the reaction rate [5,6,7,8,9,10,11,12]. In this work, the Kissinger equation was applied to obtain the activation energy for dehydrogenation of the Mg-5Ni samples whose dehydrogenation rate equation obeys the first-order law.

A rate constant *k* is generally given by the Arrhenius equation:*k* = *Z* exp (−*E*/*RT*),(1)
where *Z* is a constant, *E* is the activation energy, *R* is the gas constant, and *T* is the temperature.

The Kissinger equation is expressed by [13]
ln(*Φ*/*T*_m_^2^) = ln (*ZR*/*E*) − (*E*/*R*) (1/*T*_m_) (2)
d(ln (*Φ*/*T*_m_^2^)/d(1/*T*_m_) = −*E*/*R*
(3)
where *Φ* = d*T*/d*t* is the heating rate, *T*_m_ is the temperature at which the reaction rate has the maximum value, and *E* is the activation energy for the reaction.

In the thermal analysis methods, *T*_m_ was obtained from differential deflection versus temperature *T* curve or heat flow versus *T* curve [14]. *T*_m_ can also be obtained from the curves which are obtained in a Sieverts-type volumetric apparatus, such as
(i)d*H*_d_/d*t* versus *t* and *T* versus *t* curves(ii)d*H*_d_/d*T* versus *T* curve, and(iii)d*T*/d*t* versus *t* and *T* versus *t* curveswhere *H*_d_ is the quantity of hydrogen released from the sample, *t* is time, d*H*_d_/d*t* is the dehydrogenated rate, d*H*_d_/d*T* is the rate of increase in *H*_d_ to *T* increase, and d*T*/d*t* is the rate of increase in *T*-to-*t* increase.

As mentioned above, *T*_m_ is the temperature at which the reaction rate has the maximum value. *T*_m_ is also the temperature at which d*H*_d_/d*T* has the maximum value (*T*_m_ for the method (ii)) because *T* is increased as *t* increases. If *T* increases slowly due to the absorption of heat by hydride decomposition, d*H*_d_/d*T* will increase more rapidly.

The difference in heat between the heat supplied by the furnace and the heat absorbed by the sample for hydride decomposition will increase the temperature of the reactor. When hydrides do not decompose, all the heat supplied by the furnace will increase the temperature of the reactor. When the reaction begins, a part of the heat will be used for hydride decomposition, leading to a slower increase in the temperature of the reactor. At the moment when all the heat supplied by the furnace is consumed for hydride decomposition, the temperature of the reactor will not increase and stay constant. This is the temperature when d*T*/d*t* is zero and when the sample has the maximum hydride decomposition rate. Thus, the temperature at which d*T*/d*t* is zero in *T* versus *t* curves is *T*_m_ [*T*_m_ for the method (iii)].

In our previous work [15], we showed that the dehydrogenation rate equation of Mg-5Ni samples obeyed the first-order law, and the Kissinger equation could thus be used to determine the activation energy, and the first method above was used to obtain *T*_m_. An X-ray diffractogram of Mg-5Ni dehydrogenated at the eighth cycle showed that the sample contained Mg and small amounts of β-MgH_2_, Mg_2_Ni, and MgO. Neglecting the quantity of MgO, the composition of the sample was 97.8 mol% Mg + 2.2 mol% Mg_2_Ni. Mg and Mg_2_Ni absorb hydrogen and then release hydrogen during hydrogenation–dehydrogenation cycling.

In the present work, we used the above three methods to obtain *T*_m_. The activation energy for dehydrogenation was then calculated by using *T*_m,_ which fits the Kissinger equation best. We decided which method was the best one to calculate the activation energy for dehydrogenation.

## 2. Materials and Methods

Mg (−20 + 100 mesh, 99.8% (metals basis), Alfa Aesar, Haverhill, MA, USA) and Ni (Nickel powder APS, Singapore, 2.2–3.0 μm, purity 99.9% metal basis, C typically < 0.1%, Alfa Aesar) were used as starting materials.

For the synthesis of Mg-5Ni, milling was carried out in a planetary ball mill (Planetary Mono Mill; Pulverisette 6, Fritsch, Idar-Oberstein, Germany). A mixture of Mg and Ni at the weight ratio of 95: 5 (total weight = 8 g) was mixed in a hermetically sealed stainless-steel container (with 105 hardened steel balls, total weight = 360 g). The sample-to-ball weight ratio was 1: 45. All sample handling was performed in a glove box under an Ar atmosphere in order to minimize oxidation. The mill container (volume of 250 mL) was then filled with high-purity hydrogen gas (about 12 bar), which was refilled every 2 h. Milling was performed at the disc revolution speed of 400 rpm for 6 h.

Hydrogenation and dehydrogenation of the Mg-5Ni samples were performed in a Sieverts-type volumetric apparatus, which was similar to the one described previously [16]. We designed and constructed this apparatus in our laboratories. The Sieverts-type volumetric apparatus consisted of a reactor part and hydrogen supplying part having a known standard volume. Each part was connected to a pressure transducer. The temperature of the reactor was controlled by a programmed temperature controller. Between the reactor part and the hydrogen supplying part, a back-pressure regulator, which controlled the pressure of the reactor, was installed.

The reactor for loading the sample was in a shape of a cylinder made of S316 stainless steel with an outer diameter of 3/8” (9.5 mm) and a length of 130 mm. The amount of the used sample was 0.5 g. The conductivity of S316 stainless steel is 18.9 W/m K. The sample was heated in a vertical box furnace with a block off for heat leak by a 6T (3T + 3T) silicon pad at the entrance. The vertical box furnace was made with thermal insulation lightweight refractory bricks with ceramic fiber tube insulation in the center position. A filter was installed in the upper part of the reactor to prevent sample loss or damage to the valve parts due to the backflow of the sample in the vacuum process. Type K thermocouple probe of 1/8” diameter with stainless steel sheath was contacted along the cylinder wall of the reactor. The type K thermocouple can measure temperatures from 273 K to 1343 K. Digital PID programmable (5Pattern (9 steps/pattern)) temperature controller was used for temperature control of the furnace.

Before dehydrogenation measurements, Mg-5Ni samples were hydrogenated at 593 K under 12 bar H_2_ for 60 min. The samples were cooled to room temperature in the furnace.

For dehydrogenation measurements of the hydrogenated Mg-5Ni, the variation (increase) of the hydrogen pressure in the hydrogen supplying part, including the standard volume, was recorded as a function of time. This made us obtain curves of the quantity of hydrogen released from the sample *H*_d_ versus time *t* by using the sample weight, the volume of the hydrogen supplying part, and the room temperature. Ideal gas law was used to calculate *H*_d_ values.

The quantity of the hydrogen released under 1.0 bar H_2_ and the temperature of the reactor were measured as a function of time as the sample was heated at the heating rates (*Φ*) of 3, 6, 9, 12, and 15 K/min, respectively. The upper limit of temperature was set as 673 K.

## 3. Results

Figure 1 shows micrographs obtained by SEM (scanning electron microscopy) (JSM-5900, JEOL Ltd., Tokyo, Japan) at different magnifications of Mg-5Ni dehydrogenated at the eighth cycle [15]. Shapes of particles are irregular, and particle size is not homogeneous. Particles are agglomerated. Quite small particles are on the surfaces of agglomerates.

Figure 2 shows the variations of *H*_d_ and temperature *T* with *t* under 1.0 bar H_2_ for hydrogenated Mg-5Ni. The sample was heated with a heating rate of 6 K/min. The sample begins to release hydrogen after about 3330 s, releases hydrogen most rapidly after about 2510 s, and releases hydrogen very slowly after about 4290 s. As time elapses, the temperature increases slowly and then more rapidly. Thereafter, the temperature increases less rapidly past a point of inflection and increases more rapidly again. Finally, the temperature increases less rapidly.

The variations of *H*_d_ and temperature *T* with *t* under 1.0 bar H_2_ for hydrogenated Mg-5Ni are shown in Figure 3. The sample was heated with a heating rate of 12 K/min. The sample begins to release hydrogen after about 1930 s, releases quite rapidly from about 2060 s to 2190 s, and releases hydrogen very slowly after about 2490 s. With time, the temperature increases slowly and then more rapidly. Subsequently, the temperature passes a point of inflection, increases more rapidly again, and increases less rapidly in the end.

Figure 4 shows variations with *t* of *T* and d*H*_d_/d*t* under 1.0 bar H_2_ for hydrogenated Mg-5Ni. The sample was heated with a heating rate of 6 K/min. The temperature increases as time elapses, and a plateau appears after about 3640 s. After about 4632 s, the temperature increases very slowly. d*H*_d_/d*t* begins to increase after about 3429 s (584.05 K) and becomes very low after about 4226 s (663.05 K). d*H*_d_/d*t* is the highest after about 3760 s (601.25 K).

Figure 5 shows variations with *t* of *T* and d*H*_d_/d*t* under 1.0 bar H_2_ for hydrogenated Mg-5Ni. The sample was heated with a heating rate of 12 K/min. The temperature increases as time elapses, and a plateau appears after about 2101 s. After 2739 s, the temperature increases very slowly. d*H*_d_/d*t* begins to increase after about 1949 s (590.25 K) and becomes very low after about 2470 s (663.05 K). d*H*_d_/d*t* is the highest after about 2105 s (605.95 K).

From of d*H*_d_/d*t* versus *t* curves and *T* versus *t* curves in such as Figure 4 and Figure 5, *T*_m_s were obtained. Table 1 shows the variations in *T*_m_, ln (*Φ*/*T*_m_^2^), and 1/*T*_m_, with heating rate *Φ* under 1.0 bar H_2_ for hydrogenated Mg–5Ni, obtained from the data of d*H*_d_/d*t* versus *t* curves and *T* versus *t* curves.

Figure 6 shows the plot of ln (*Φ*/*T*_m_^2^) versus 1/*T*_m_, obtained from the data of d*H*_d_/d*t* versus *t* curves and *T* versus *t* curves. The linearity of the plot is very poor (correlation coefficient: −0.936). Assuming that the plot had good linearity, the activation energy for dehydrogenation, *E*, was calculated as 206 KJ/mol from this plot.

Figure 7 shows the variations with temperature, *T*, of *H*_d_ and the ratio of *H*_d_ change to *T* change, d*H*_d_/d*T*, under 1.0 bar H_2_ for hydrogenated Mg-5Ni. The sample was heated with a heating rate of 6 K/min. The d*H*_d_/d*T* begins to increase after about 573 K, is the highest at about 595 K, and becomes very low after about 663 K.

Figure 8 shows the variations with temperature, *T*, of *H*_d_ and d*H*_d_/d*T* under 1.0 bar H_2_ for hydrogenated Mg-5Ni. The sample was heated with a heating rate of 12 K/min. The d*H*_d_/d*T* begins to increase after about 590 K, is the highest at about 605 K, and becomes very low after about 663 K.

From d*H*_d_/d*T* versus *T* curve in such as Figure 7 and Figure 8, *T*_m_s were obtained. Table 2 shows the variations in *T*_m_, ln (*Φ*/*T*_m_^2^), and 1/*T*_m_, with heating rate *Φ* under 1.0 bar H_2_ for hydrogenated Mg–5Ni, obtained from the data of d*H*_d_/d*T* versus *T* curves.

Figure 9 shows the plot of ln (*Φ*/*T*_m_^2^) versus 1/*T*_m_, obtained from the data of d*H*_d_/d*T* versus *T* curves. The linearity of the plot is poor (correlation coefficient: −0.965). Assuming that the plot had good linearity, the activation energy for dehydrogenation, E, was calculated as 198 KJ/mol.

Figure 10 shows the *T* versus *t* and d*T*/d*t* versus *t* curves for hydrogenated Mg-5Ni when heated with a heating rate of 6 K/min. As time elapses, *T* increases slowly in the beginning and then increases almost linearly. At about 594 K, a point of inflection appears and then increases almost linearly. After about 675 K, the temperature increases very slowly. As time elapses, d*T*/d*t* increases almost linearly in the beginning and then stays at a constant value roughly. Thereafter, d*T*/d*t* decreases become the lowest at 594.35 K and again increases. After about 625 K, d*T*/d*t* decreases. The decrease in d*T*/d*t* after about 625 K is due to a slower increase in temperature (according to the smaller power supply of the temperature controller) as the temperature reaches the upper limit set. The upper limit of temperature set by the thermal program in the temperature controller was 673 K. The fluctuation of the d*T*/d*t* versus *t* curve is believed to appear because the *T* versus *t* curve has a shape like stairs with different slopes when it is observed on a very large scale.

The *T* versus *t* and d*T*/d*t* versus *t* curves for hydrogenated Mg-5Ni when heated with a heating rate of 12 K/min are shown in Figure 11. As time elapses, *T* increases slowly in the beginning and then increases almost linearly. At about 604 K, a point of inflection appears and then increases almost linearly. After about 675 K, the temperature increases very slowly. As time elapses, d*T*/d*t* increases almost linearly in the beginning and then increases slowly. Thereafter, d*T*/d*t* decreases, becomes the lowest at 604.15 K, and increases again. After about 635 K, d*T*/d*t* decreases.

From d*T*/d*t* versus *t* curve in such as Figure 10 and Figure 11, *T*_m_s were obtained. Table 3 shows the variations in *T*_m_, ln (*Φ*/*T*_m_^2^), and 1/*T*_m_, with heating rate *Φ* under 1.0 bar H_2_ for hydrogenated Mg–5Ni, obtained from the data of *T* versus *t* and d*T*/d*t* versus *t* curves.

Figure 12 shows the plot of ln (*Φ*/*T*_m_^2^) versus 1/*T*_m_, obtained from the data of *T* versus *t* and d*T*/d*t* versus *t* curves. The linearity of the plot is very good (correlation coefficient: −0.996). From this plot, the activation energy for dehydrogenation, *E*, was calculated as 210 ± 15 KJ/mol. The margin of error was relatively small.

Among the plots of ln (*Φ*/*T*_m_^2^) versus 1/*T*_m_ using *T*_m_ obtained from three methods, the plot using *T*_m_ obtained from the third method (using d*T*/d*t* versus *t* and *T* versus *t* curves) exhibits the best linearity, showing that the third method is the best one to obtain the activation energy for dehydrogenation of the hydrogenated Mg-5Ni samples.

Comparing the equation obtained from Figure 12 with Equation (2), *E* was 210 ± 15 kJ/mole, and *Z* was 1.37 × 10^18^. The rate constant, *k*, can thus be expressed by Equation (4):*k* = (1.37 × 10^18^) exp [−210 ± 15 (kJ/mole)/*RT*].(4)

## 4. Discussion

In our laboratory, many works were performed to increase the hydrogenation and dehydrogenation rates of Mg by adding metallic elements or compounds [17,18,19,20,21,22], which forms defects, clean surfaces, and decreases the particle size. In this work, 5 wt% Ni-added Mg samples, Mg-5Ni samples, were prepared.

Because the Kissinger equation was derived for the thermal decomposition on heating which obeyed a first-order rate law [13,14], the obedience of the first-order rate law by the reaction must be checked before applying data to the Kissinger equation to obtain the activation energy.

In the *T* versus *t* curves of Figure 2, Figure 3, Figure 4, Figure 5, Figure 10 and Figure 11, a point of inflection appears. The temperature at this point is *T*_m_. At the end of the *T* versus *t* curves, the temperature increases less rapidly because, even though the dehydrogenation reaction stops, less heat is supplied by the furnace since the temperature limit is set as 673 K in the program of the temperature controller.

In the d*T*/d*t* versus *t* curves of Figure 10 and Figure 11, d*T*/d*t* decreases after about 625 K and 635 K, respectively. This decrease in d*T*/d*t* is due to a slower increase in temperature (according to the smaller power supply of the temperature controller) as the temperature reaches the upper limit set. The upper limit of temperature set by the program in the temperature controller was 673 K.

As the temperature of the reactor in the Sieverts-type apparatus increases monotonically, d*H*_d_/d*T* has a maximum value when the dehydrogenation rate is the highest. The temperature at which d*H*_d_/d*T* has a maximum value is thus *T*_m_. At the moment when all the heat supplied by the furnace is consumed for dehydrogenation, the temperature of the reactor will not increase and stay constant. This is the temperature when d*T*/d*t* is zero and when the sample has the maximum dehydrogenation rate. Thus, the temperature at which d*T*/d*t* is zero in *T* versus *t* curves is *T*_m_.

After d*T*/d*t* is zero in the *T* versus *t* curves, the heat supplied by the furnace will be larger than the heat absorbed by the sample for dehydrogenation, and the temperature of the reactor will increase.

To search for *T*_m_, the first and second methods use *H*_d_ and *T* values, whereas the third method only uses the *T* value. The sample is in the form of powder. Heat transfer among the particles is poor. Therefore, there can be a delayed time in heat transfer from the sample to the thermocouple. This delayed time is believed to have led to incorrect reading of the value of *T*_m_ (because *T* is read from *T* versus *t* curves) in the first and second methods. Meanwhile, even though there is a delayed time in heat transfer, no error in the calculation may be involved in the third method because similar delayed times are expected for all different heating rates. In addition, the errors involved in calculating *H*_d_ are those from the variation of room temperature, the thermocouple for the reactor, pressure transducers, and the distribution of temperature in the Sieverts-type apparatus. On the other hand, the error involved in calculating *T* is only the thermocouple for the reactor. These two points are believed to have led to better results (a better fitting to the Kissinger equation) of the third method than those of the first and second methods.

Zhao-Karger et al. [5] carried out thermal analyses for MgH_2_ decomposition of the commercial MgH_2_ using high-pressure DSC and simultaneous TGA-DSC-MS (mass spectrometry). The Kissinger plot for the data from these thermal analyses showed that the activation energy for this reaction was 195 KJ/mol. Sabitu and Goudy [6] performed TG/DTA for the MgH_2_ decomposition of an MgH_2_ sample (without oxide). They reported that the Kissinger plot, obtained from the TG/DTA data, showed that the activation energy for MgH_2_ decomposition of MgH_2_ (without oxide) was 174 KJ/mol. Xiao et al. [7] performed thermal analysis using TG-DTA for MgH_2_ decomposition of an as-received MgH_2_. The Kissinger plot showed that the activation energy for this reaction was 213 KJ/mol. Campostrini et al. [8] investigated the thermal desorption of hydrogen from commercial MgH_2_ powders by coupled thermogravimetry and mass spectroscopy (TG–MS). They reported that considering temperatures corresponding to iso-conversion points of transformed fraction = 0.25 and calculating the relative values of the reaction rate, the activation energy of 240 kJ/mol for the hydrogen release from commercial MgH_2_ was obtained via the Friedman method [8]. In this work, the activation energy for the hydrogenated Mg-5Ni samples was obtained from the data collected in a Sieverts-type volumetric apparatus. Obtaining data in a Sieverts-type volumetric apparatus, compared to the thermal analysis methods, has the advantage that data can be obtained in the same apparatus where samples were activated and hydrogenated. The Mg-5Ni sample is composed of 97.8 mol% Mg and 2.2 mol% Mg_2_Ni. Because it was hard to find the results of the samples with compositions similar to Mg-5Ni and the mol% of Mg_2_Ni is relatively small, we compared the results of this work with those of MgH_2_. The activation energy for dehydrogenation (210 kJ/mole) of the hydrogenated Mg-5Ni samples obtained in the present work is larger than that of Zhao-Karger et al. [5] for the hydrogen release of the commercial MgH_2_ (195 kJ/mol) and that of Sabitu and Goudy [6] for MgH_2_ decomposition of MgH_2_ without oxide (174 kJ/mol), but smaller than that of Campostrini et al. [8] for the hydrogen release of the commercial MgH_2_ powders (240 kJ/mol). Our result has a value very similar to that of Xiao et al. [7] for the MgH_2_ decomposition of an as-received MgH_2_ (213 KJ/mol).

Han et al. [9,10] reported that the apparent activation energy for hydride decomposition can be obtained from ln (*b/T*_p_^2^) versus 1/*T*_p_ plot, where *b* is the heating rate and *T*_p_ is the peak temperature of the desorption rate versus temperature curve. This plot is very similar to that to obtain the activation energy using the Kissinger equation, but the apparent activation energy for hydride decomposition corresponded to the activation energy of hydride decomposition plus the partial molar heat of solution of hydrogen minus half the reaction heat for the reaction from a metal-hydrogen solid solution to a hydride. They derived the equation based on the continuous moving boundary model, assuming that the particles were spherical with a uniform diameter and that the rate-controlling step for hydride decomposition was an interfacial chemical reaction, the rate equation of which obeys the first-order law.

Now, we mention the results of our previous work [3]. Figure 6 shows the plot of ln (*Φ*/*T*_m_^2^) versus 1/*T*_m_, obtained from the data of d*H*_d_/d*t* versus *t* curves and *T* versus *t* curves. The linearity of the plot is very poor (correlation coefficient: −0.936). From this plot, the activation energy for dehydrogenation, *E*, was calculated as 206 KJ/mol by assuming that the linearity of the plot was good. In our previous work [3], the obtained activation energy for dehydrogenation was 174 KJ/mol, which is smaller than that obtained from Figure 6. In that work [3], *T*_m_ was obtained, assuming that *T*_m_ was the temperature at the time corresponding to the center of the asymmetric peak. This assumption is thought to have led to a different result, which was shown to be incorrect by the result of the present work.

In future work, by improving the contact of the thermocouple, the accuracy of the temperature controller in the furnace, and maintaining the room temperature constant, more accurate temperature variation with time will be obtained, leading to more accurate results of the obtained activation energy. By performing research similar to the present work for other samples prepared by different procedures and/or with different compositions, the usefulness of the three methods suggested in this work will be checked and discussed.

## 5. Conclusions

We showed that the dehydrogenation rate equation of hydrogenated Mg-5Ni samples (composition: 97.8 mol% Mg + 2.2 mol% Mg_2_Ni) obeyed the first-order law, and the Kissinger equation could thus be used to determine the activation energy in our previous work. The present work suggested three methods to obtain the activation energy *E* of dehydrogenation (on heating at constant rates) from data obtained in a Sieverts-type volumetric apparatus by applying the Kissinger equation. *T*_m_ (the temperature at which the dehydrogenation rate has the maximum value) can be obtained from (i) d*H*_d_/d*t* versus *t* and *T* versus *t* curves, (ii) d*H*_d_/d*T* versus *T* curve, and (iii) d*T*/d*t* versus *t* and *T* versus *t* curves. Among the plots of ln (*Φ*/*T*_m_^2^) versus 1/*T*_m_ using *T*_m_ obtained from three methods, the plot using *T*_m_ obtained from the third method (using d*T*/d*t* versus *t* and *T* versus *t* curves) exhibited the best linearity, the margin of error being relatively small. The activation for the dehydrogenation of hydrogenated Mg-5Ni samples calculated from the third method was 210 ± 15 kJ/mole. These three methods can be used to calculate the activation energy from data of the constant-heating rate reactions obtained in a Sieverts-type apparatus. Among these three methods, it is believed that the third method is the best because a delayed time in heat transfer from the sample to the thermocouple has similar effects on the value of *T*_m_ and the third method involves the least error thanks to the measurement of only temperature.

## Figures and Tables

**Figure 1 micromachines-13-01809-f001:**
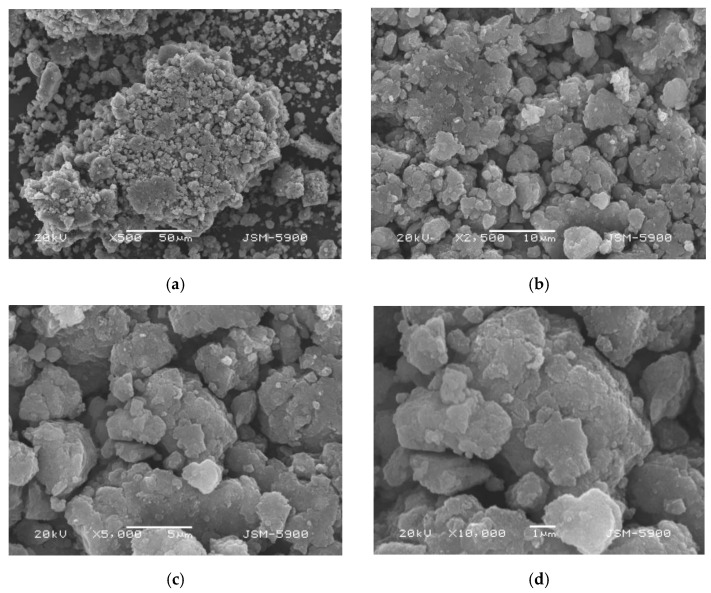
Micrographs captured by SEM at various magnifications ((**a**) ×500, (**b**) ×2500, (**c**) ×5000, and (**d**) ×10,000) of Mg-5Ni dehydrogenated at the eighth cycle.

**Figure 2 micromachines-13-01809-f002:**
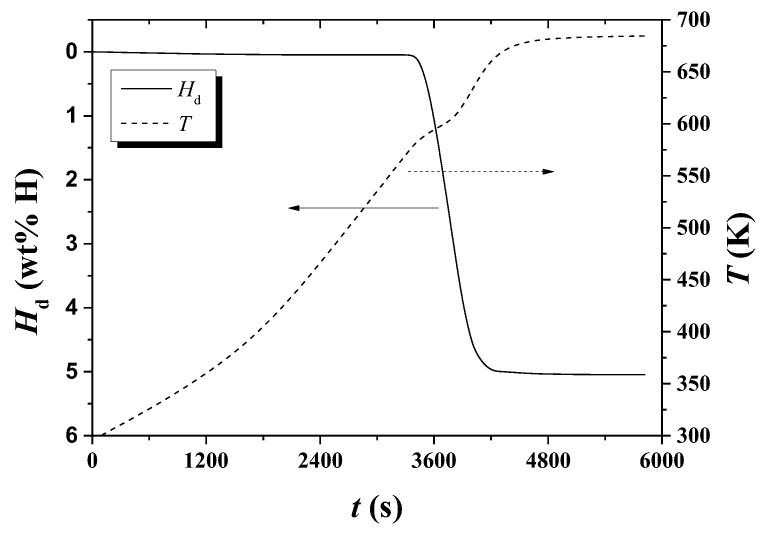
Variations of *H*_d_ and temperature *T* with time *t* under 1.0 bar H_2_ for hydrogenated Mg-5Ni. The sample was heated with a heating rate of 6 K/min.

**Figure 3 micromachines-13-01809-f003:**
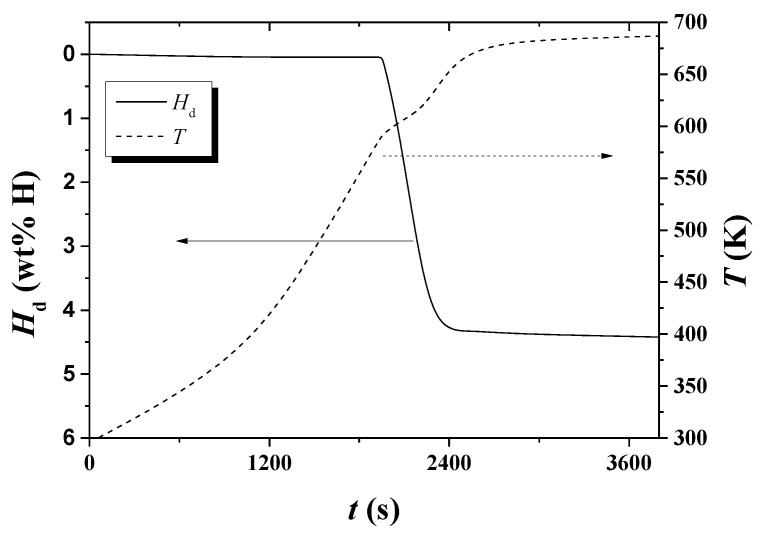
Variations of *H*_d_ and *T* with *t* under 1.0 bar H_2_ for hydrogenated Mg-5Ni. The sample was heated with a heating rate of 12 K/min.

**Figure 4 micromachines-13-01809-f004:**
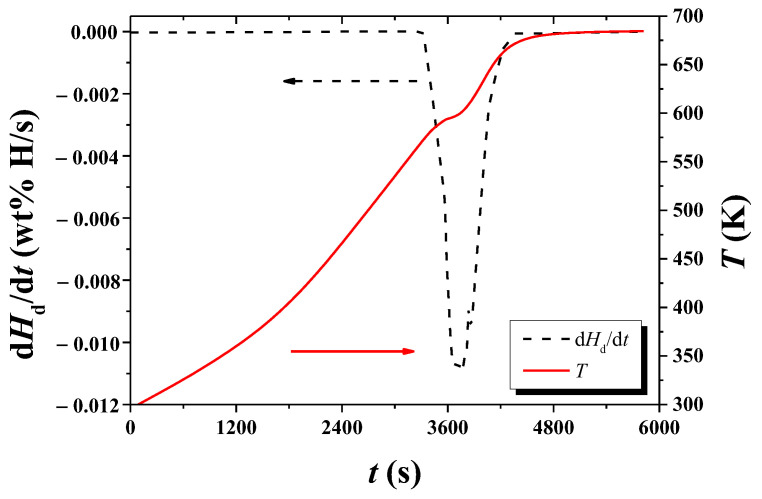
Variations with *t* of *T* and d*H*_d_/d*t* under 1.0 bar H_2_ for hydrogenated Mg-5Ni. The sample was heated with a heating rate of 6 K/min.

**Figure 5 micromachines-13-01809-f005:**
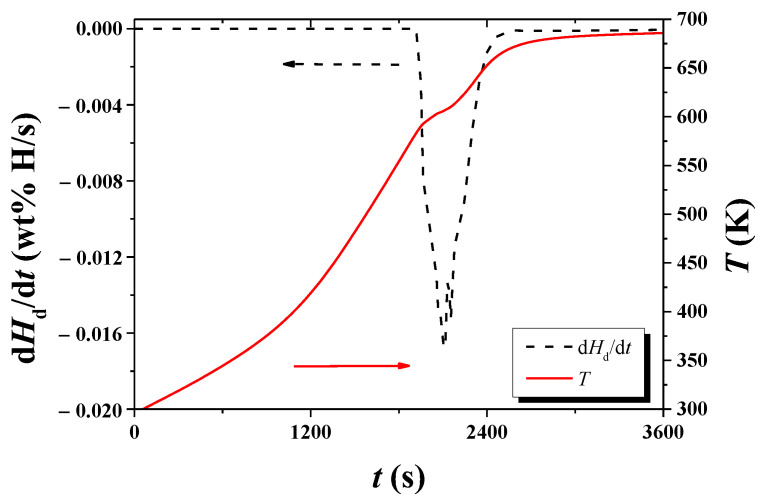
Variations with *t* of *T* and d*H*_d_/d*t* under 1.0 bar H_2_ for hydrogenated Mg-5Ni. The sample was heated with a heating rate of 12 K/min.

**Figure 6 micromachines-13-01809-f006:**
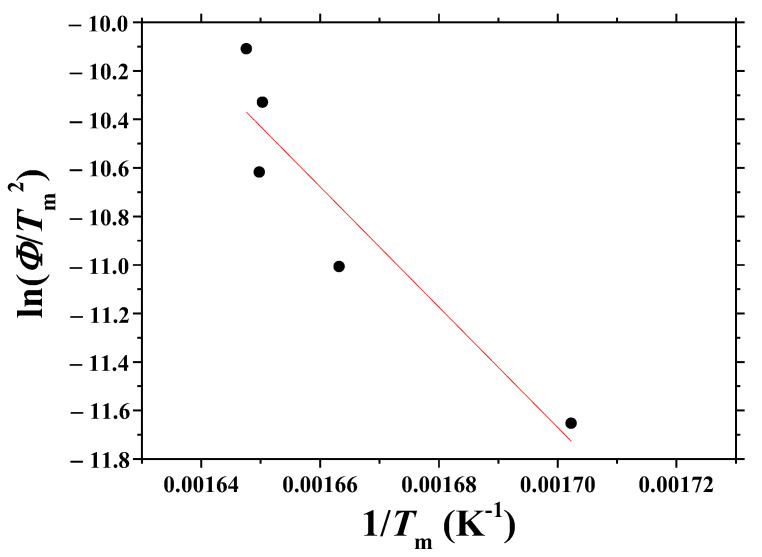
Plot of ln (*Φ*/*T*_m_^2^) versus 1/*T*_m_, obtained from the data of d*H*_d_/d*t* versus *t* curves and *T* versus *t* curves.

**Figure 7 micromachines-13-01809-f007:**
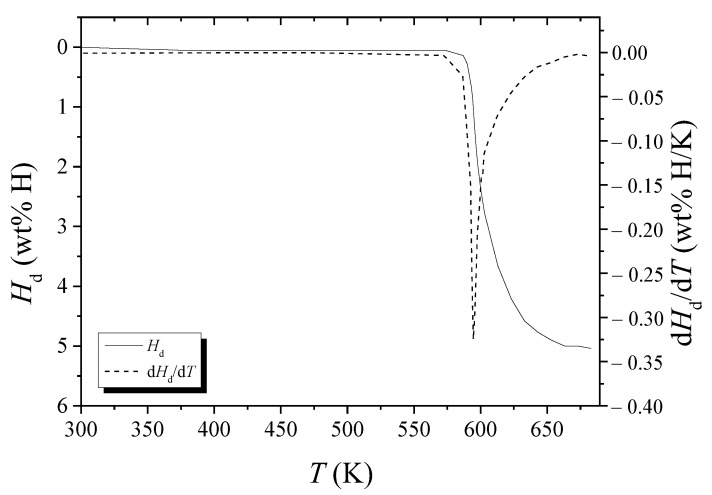
Desorbed hydrogen quantity *H*_d_ versus *T* curve and d*H*_d_/d*T* versus *T* curve for hydrogenated Mg-5Ni when heated with a heating rate of 6 K/min.

**Figure 8 micromachines-13-01809-f008:**
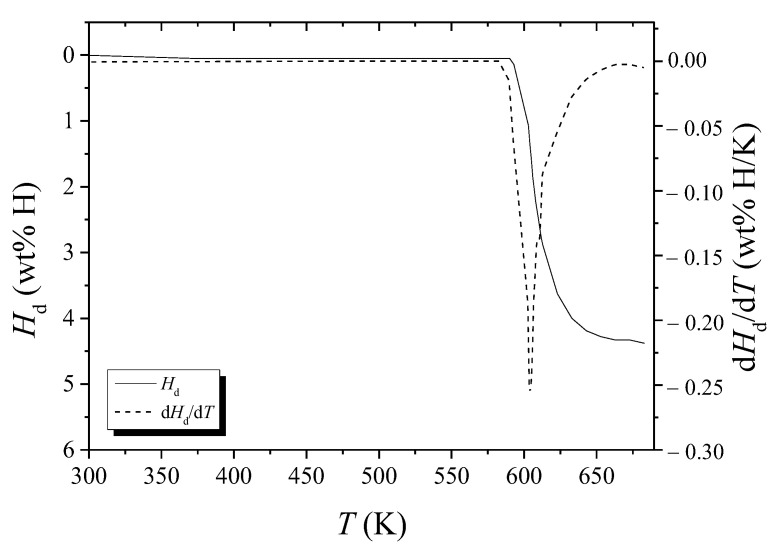
Desorbed hydrogen quantity *H*_d_ versus *T* curve and d*H*_d_/d*T* versus *T* curve for hydrogenated Mg-5Ni when heated with a heating rate of 12 K/min.

**Figure 9 micromachines-13-01809-f009:**
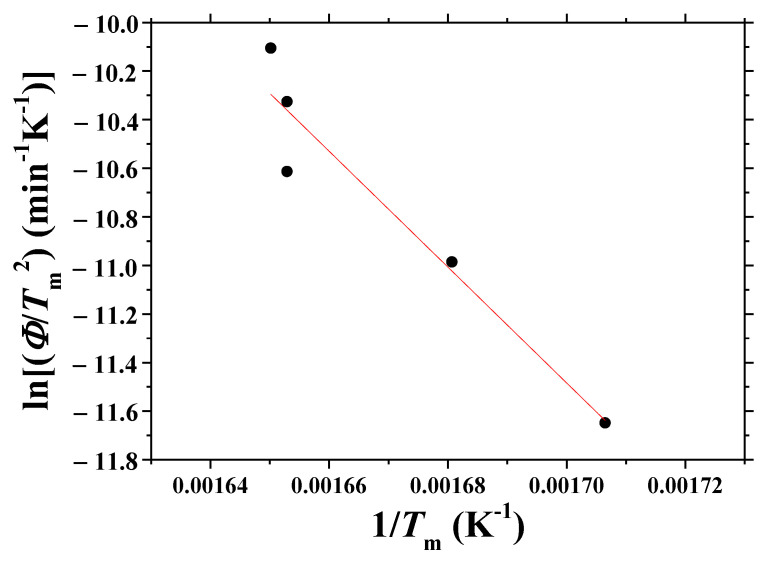
Plot of ln (*Φ*/*T*_m_^2^) versus 1/*T*_m_, obtained from the data of d*H*_d_/d*T* versus *T* curves.

**Figure 10 micromachines-13-01809-f010:**
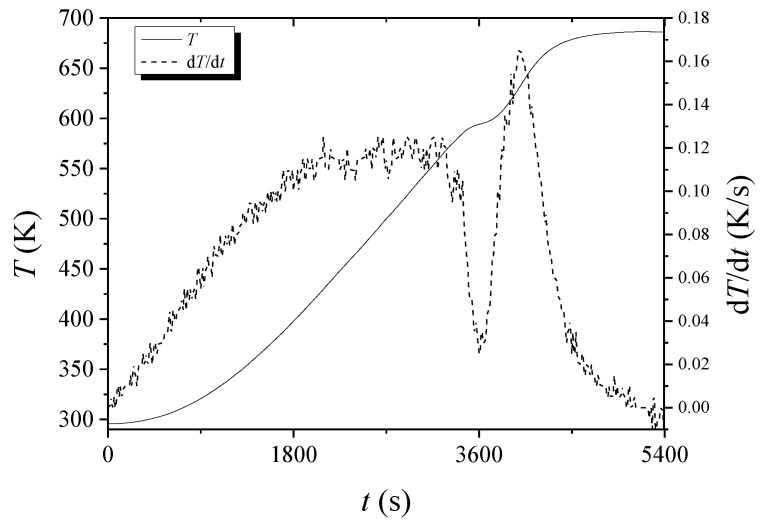
*T* versus *t* and d*T*/d*t* versus *t* curves for hydrogenated Mg-5Ni when heated with a heating rate of 6 K/min.

**Figure 11 micromachines-13-01809-f011:**
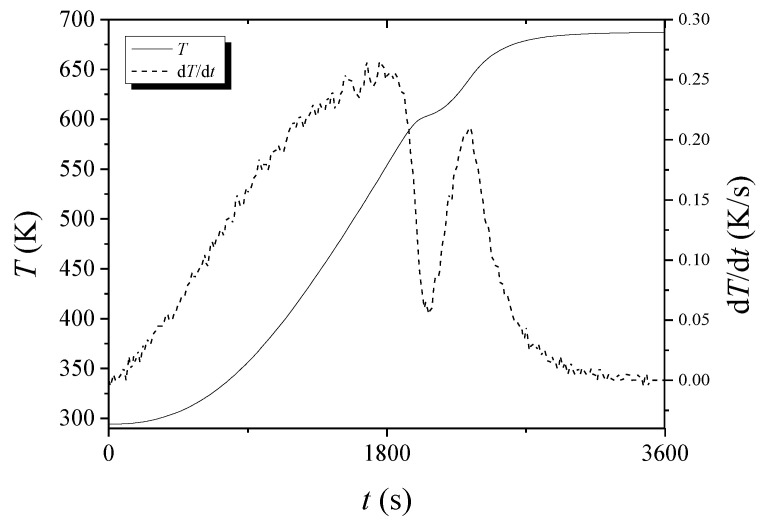
*T* versus *t* and d*T*/d*t* versus *t* curves for hydrogenated Mg-5Ni when heated with a heating rate of 12 K/min.

**Figure 12 micromachines-13-01809-f012:**
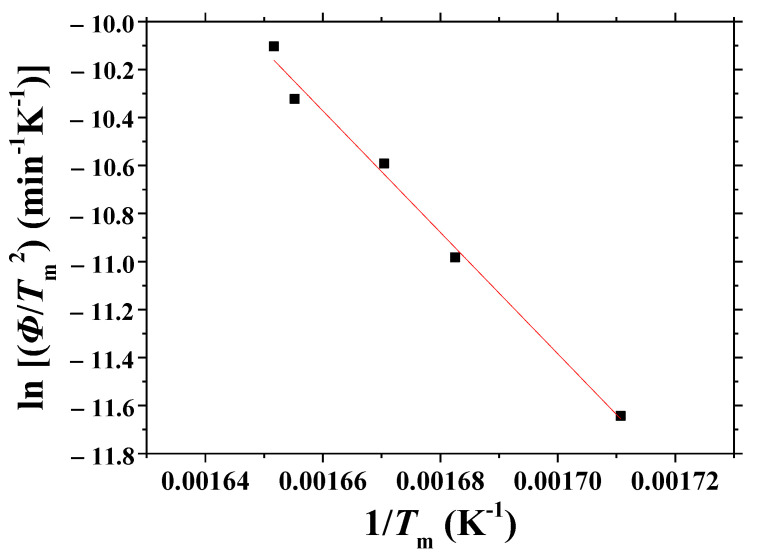
Plot of ln (*Φ*/*T*_m_^2^) versus 1/*T*_m_, obtained from the data of *T* versus *t* and d*T*/d*t* versus *t* curves.

**Table 1 micromachines-13-01809-t001:** Variations in *T*_m_, ln (*Φ*/*T*_m_^2^), and 1/*T*_m_, with heating rate *Φ* under 1.0 bar H_2_ for hydrogenated Mg–5Ni, obtained from the data of d*H*_d_/d*t* versus *t* curves and *T* versus *t* curves.

*Φ* (K/min)	*T*_m_ (K)	ln (*Φ*/*T*_m_^2^)	1/*T*_m_
3	587.45	−11.65296998	0.001702273
6	601.25	−11.00626217	0.001663202
9	606.15	−10.61703038	0.001649757
12	605.95	−10.3286883	0.001650301
15	606.95	−10.10884263	0.001647582

**Table 2 micromachines-13-01809-t002:** Variations in *T*_m,_ ln (*Φ*/*T*_m_^2^), and 1/*T*_m_, with heating rate *Φ* under 1.0 bar H_2_ for hydrogenated Mg–5Ni, obtained from the data of d*H*_d_/d*T* versus *T* curves.

Heating Rate *Φ* (K/min)	*T*_m_ (K)	ln (*Φ*/*T*_m_^2^)	1/*T*_m_
3	586	−11.64802729	0.001706485
6	595	−10.98536334	0.001680672
9	605	−10.61323234	0.001652893
12	605	−10.32555027	0.001652893
15	606	−10.10570977	0.001650165

**Table 3 micromachines-13-01809-t003:** Variations in *T*_m_, ln (*Φ*/*T*_m_^2^), and 1/*T*_m_, with heating rate *Φ* under 1.0 bar H_2_ for hydrogenated Mg–5Ni, obtained from the data of *T* versus *t* and d*T*/d*t* versus *t* curves.

Heating Rate *Φ* (K/min)	*T*_m_ (K)	ln (*Φ*/*T*_m_^2^)	1/*T*_m_
3	584.55	−11.64307235	0.001710718
6	594.35	−10.98317727	0.001682510
9	598.65	−10.59212966	0.001670425
12	604.15	−10.32273837	0.001655218
15	605.45	−10.10389377	0.001651664

## Data Availability

Data presented in this article are available upon request from the corresponding author.

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
