# Peer review of "Three Methods for Application of Data from a Volumetric Method to the Kissinger Equation to Obtain Activation Energy"

_micromachines, 2022, doi:10.3390/mi13111809_

Round 1
Reviewer 1 Report
In this manuscript, three Methods were adopted to obtain Activation Energy. Articles need to be reorganized, and more systematic experiments need to be contrasted to verify the accuracy of the method. The article cannot be published in the current state.
1) The presentation of the novelty of the paper is weak. This is a crucial point for a scientific publication and must emerge from the indication of the literature gaps.
2) The comments on the achieved results require the clear and deep explanation of their physical motivations. Moreover, the comparison of the presented outcomes with those of similar studies should be included in the text to corroborate the statements.
3)The references cited in the article are too few and the discussion is not in-depth enough
4) The introduction section needs to be reorganized to highlight the idea of the article.
Author Response
In this manuscript, three Methods were adopted to obtain Activation Energy. Articles need to be reorganized, and more systematic experiments need to be contrasted to verify the accuracy of the method. The article cannot be published in the current state.
1). The presentation of the novelty of the paper is weak. This is a crucial point for a scientific publication and must emerge from the indication of the literature gaps.
Response: The following was added in Introduction:
For the thermal analyses, the samples should be transferred to thermal analysis equipments after hydrogenation. To prevent the samples from being oxidized, equipments to isolate the hydrogenated samples are required. Or the volumetric hydrogenation equipments and the thermal analysis equipments should be kept in inert atmosphere simultaneously. In some institutes, analysis using thermal analysis equipments is not easy to be performed because evolved hydrogen is considered to contaminate parts of sensors for thermal analysis equipments. In the present work, we obtained the activation energy for dehydrogenation of hydrogenated Mg-5Ni samples by applying data from a volumetric method to the Kissinger equation. This method has an advantage: data can be obtained in the same apparatus where samples were activated and hydrogenated.
In order that the Kissinger equation can be used to obtain the activation energy, the reaction rate equation must obey a first-order law [13]. However, the previous works which obtained the activation energy were performed without showing the obedience of the first-order law by the reaction rate [5-12]. In this work, the Kissinger equation was applied to obtain the activation energy after showing that the hydrogenation rate equation obeys a first-order law.
2). The comments on the achieved results require the clear and deep explanation of their physical motivations. Moreover, the comparison of the presented outcomes with those of similar studies should be included in the text to corroborate the statements.
Response: We tried to explain clearly the achieved results. The presented outcomes were compared with those of MgH2: Mg-5Ni sample is composed of 97.8 mol% Mg and 2.2 mol% Mg2Ni. Because it was hard to find the results of the samples with compositions similar to Mg-5Ni, we compared the results of this work with those of MgH2. The hydrogenated sample of pure Mg releases very slowly. We added a small amount of Ni (5 wt%) to make the samples have dehydrogenation rates high enough to measure the dehydrogenation rate easily in this work.
In this work, the activation energy for the hydrogenated Mg-5Ni samples were obtained from the data collected in a Sieverts-type volumetric apparatus. Obtaining data in in a Sieverts-type volumetric apparatus, compared in the thermal analysis methods, has the advantage that data can be obtained in the same apparatus where samples were activated and hydrogenated. The activation energy for dehydrogenation (210 kJ/mole) of the hydrogenated Mg-5Ni samples obtained in the present work is larger than that of Zhao-Karger et al. [5] for the hydrogen release of the commercial MgH2 (195 kJ/mol) and that of Sabitu and Goudy [6] for MgH2 decomposition of MgH2 without oxide (174 kJ/mol), but smaller than that of Campostrini et al. [8] for the hydrogen release of the commercial MgH2 powders (240 kJ/mol). Our result has a value very similar to that of Xiao et al. [7] for MgH2 decomposition of an as-received MgH2 (213 KJ/mol).
3). The references cited in the article are too few and the discussion is not in-depth enough
Response: References were added. The discussion was corrected and added.
4). The introduction section needs to be reorganized to highlight the idea of the article.
Response: The introduction was added and reorganized:
Hydrogen has drawn attention as a next-generation energy carrier. To use hydrogen as energy carrier, hydrogen-storage materials should be developed. Metal hydrides such as hydrides of Mg-based alloys [1-4] are one of the candidates for hydrogen-storage materials.
For the thermal analyses, the samples should be transferred to thermal analysis equipments after hydrogenation. To prevent the samples from being oxidized, equipments to isolate the hydrogenated samples are required. Or the volumetric hydrogenation equipments and the thermal analysis equipments should be kept in inert atmosphere simultaneously. Analysis using thermal analysis equipments is not always easy to be performed because evolved hydrogen is considered to contaminate parts of sensors for thermal analysis equipments. In the present work, we obtained the activation energy for dehydrogenation of hydrogenated Mg-5Ni samples by applying data from a volumetric method to the Kissinger equation. This method has an advantage: data can be obtained in the same apparatus where samples were activated and hydrogenated.
In order that the Kissinger equation can be used to obtain the activation energy, the reaction rate equation must obey a first-order law [13]. However, the previous works which obtained the activation energy without showing the obedience of the first-order law by the reaction rate [5-12]. In this work, the Kissinger equation was applied to obtain the activation energy for dehydrogenation of the Mg-5Ni samples whose dehydrogenation rate equation obeys a first-order law.
Thank you very much for your detailed review and comments.

Reviewer 2 Report
I get the point to try using a Siverts type apparatus to obtain data typically obtained from DSC measurements. The principle of application of the Kissinger method is sound; however, there is a lack of experimental good performance.
In section 2. Materials and Methods, a better explanation of the operation/ use/ data collection of the Sieverts-type apparatus must be presented. Also, details of the construction, instrumentation, and programming of the Sieverts apparatus can help explain the results.
Results such as those presented in Fig. 6 demonstrated poor experimental control and selection of experimental parameters. Such results would suggest any reader "don't use the proposed method". Results such as Fig. 12 are expected and suggest to any potential reader "yes, use the proposed method". Thus, a revision and repetition of experiments are needed.
Discussion is a bit pointless because i) the control type of the heating of the furnace is not explained in the experimental section, i. e. it does not explain properly the deviation from linear expectations. ii) It is pointless to compare to MgH2 because the material is Mg-5Ni (where some formation of Mg2Ni/Mg2NiH4 would be expected) and the processing of materials can play an important role.
Additionally, some parts are difficult to read, for example, the Abstract (lines 15-21).
Author Response
I get the point to try using a Siverts type apparatus to obtain data typically obtained from DSC measurements. The principle of application of the Kissinger method is sound; however, there is a lack of experimental good performance.
Response: Among the plots of ln (Φ/Tm2) versus 1/Tm using Tm obtained from the three methods, the plot using Tm obtained from the third method (Fig, 12. using dT/dt versus t and T versus t curves) exhibits the best linearity, showing that the third method is quite a good one to obtain the activation energy for dehydrogenation of the hydrogenated Mg-5Ni samples.
In section 2. Materials and Methods, a better explanation of the operation/ use/ data collection of the Sieverts-type apparatus must be presented. Also, details of the construction, instrumentation, and programming of the Sieverts apparatus can help explain the results.
Response: The Sieverts-type volumetric apparatus consisted of a reactor part and a hydrogen supplying part having a known standard volume. Each part was connected to a pressure transducer. The temperature of the reactor was controlled by a programmed temperature controller. Between the reactor part and the hydrogen supplying part, a back-pressure regulator, which controlled the pressure of the reactor, was installed. The variation of the hydrogen pressure in the hydrogen supplying part (including the standard volume) was recorded as a function of time. This made us to obtain curves of the quantity of hydrogen released from the sample Hd versus time t. Series of the values of Hd was calculated from the program using sample weight, the volume of the hydrogen supplying part, and room temperature.
Results such as those presented in Fig. 6 demonstrated poor experimental control and selection of experimental parameters. Such results would suggest any reader "don't use the proposed method". Results such as Fig. 12 are expected and suggest to any potential reader "yes, use the proposed method". Thus, a revision and repetition of experiments are needed.
Response: You are right.
Among the plots of ln (Φ/Tm2) versus 1/Tm using Tm obtained from three methods, the plot using Tm obtained from the third method (Fig, 12. using dT/dt versus t and T versus t curves) exhibits the best linearity, showing that the third method is the best one to obtain the activation energy for dehydrogenation of the hydrogenated Mg-5Ni samples.
To search for Tm, the first and second methods use Hd and T values whereas the third method only uses T value. The sample is in a form of powder. Heat transfer among the particles is poor. So, there can be a delayed time in heat transfer from the sample to the thermocouple. This delayed time is believed to have led to incorrect reading of the value of Tm (because T is read from T versus t curves) in the first and second methods. Meanwhile, even though there is the delayed time in heat transfer, no error in calculation may be involved in the third method because similar delayed times are expected for all different heating rates. In addition, the errors involved to calculate Hd are those from room temperature, a thermocouple for the reactor, pressure transducers, and the distribution of temperature in the Sieverts-type apparatus. On the other hand, the error involved to calculate T is only a thermocouple for the reactor. These two points are believed to have led to better results (a better fitting to the Kissinger equation) of the third method than those of the first and second methods.
Excuse us, but it is not easy to repeat the experiments in a short period to prepare the revised manuscript. The apparatus is under repair due to leak problem and change of parts. Please allow us to obtain data of repeated experiments in the future.
Discussion is a bit pointless because i) the control type of the heating of the furnace is not explained in the experimental section, i. e. it does not explain properly the deviation from linear expectations.
Response:
The following was added in Materials and Methods section: Digital PID programmable [5Pattern (9 steps /pattern)] temperature controller was used for temperature control of the furnace. The sample was heated in a vertical box furnace with block off for heat leak by 6T (3T+3T) silicon pad in entrance. The vertical box furnace was made by thermal insulation lightweight refractory bricks with ceramic fiber tube insulation in center position.
While dehydrogenation does not occur, the temperature of the reactor in the Sieverts-type apparatus increases linearly.
When the temperature of the reactor in the Sieverts-type apparatus increases monotonically, dHd/dT has a maximum value when the dehydrogenation rate is the highest. The temperature at which dHd/dT has a maximum value is thus Tm. The difference of heat between the heat supplied by the furnace and the heat absorbed by the sample for dehydrogenation will increase the temperature of the reactor. At the moment when all the heat supplied by the furnace is consumed for dehydrogenation, the temperature of the reactor will not increase and stay constant. This is the temperature when dT/dt is zero and when the sample has the maximum dehydrogenation rate. Thus, the temperature at which dT/dt is zero in T versus t curves is Tm. After dT/dt is zero in T versus t curves, the heat supplied by the furnace will be larger than the heat absorbed by the sample for dehydrogenation and the temperature of the reactor will increase.
- ii) It is pointless to compare to MgH2 because the material is Mg-5Ni (where some formation of Mg2Ni/Mg2NiH4 would be expected) and the processing of materials can play an important role.
Response: Mg-5Ni sample is composed of 97.8 mol% Mg and 2.2 mol% Mg2Ni. Because it was hard to find the results of the samples with compositions similar to Mg-5Ni, we compared the results of this work with those of MgH2. The hydrogenated sample of pure Mg releases very slowly. We added a small amount of Ni (5 wt%) to make the samples have dehydrogenation rates high enough to measure the dehydrogenation rate easily in this work.
Additionally, some parts are difficult to read, for example, the Abstract (lines 15-21).
Response: Some parts of the Abstract were corrected.
We appreciate you reviewing in detail and giving comments.

Reviewer 3 Report
This paper is devoted to the features of the dehydrogenation of home-made Mg-5Ni alloy in the Sieverts-type volumetric apparatus and the calculation of the kinetics of this process using the well-known Kissinger equation. The authors demonstrate a new empirical approach to estimation of Tm (temperature at which the reaction rate is highest) and to more accurate calculation of activation energy analyzing the function of dT/dt (T – temperature, t-time of reaction). I think that these results should be published in the Micromachines journal. The Sieverts-type reactors and the Kissinger equation are used quite often for different solid-phase dehydrogenation processes. These results are helpful for the readers. But major revision is necessary.
Major remarks
1 The paper contains only 8 references, 2 of them are the works of the authors. The content of the Introduction does not show the problem being solved by the authors. Some of this information is presented in the Discussion section at the end of the paper. The Kissinger kinetic model is the most used in studying the kinetics of the dehydrogenation of MgH2 and Mg-based alloys. What is the problem in such studies? Is this a scatter of the data in activation energy? Is this a big error in linear approximation? Is this an effect of the setup and experimental details on the results? Now, the information that the authors present in the Introduction is repeated several more times at the discussion of the results obtained. This part may be shortened as it contains generally understandable conclusions. I strongly advise the authors to rewrite the Introduction and expand the number of citations. Defining the requirements for the using the Kissinger equation and the already proposed ways to improve the calculation of kinetics using this model are useful.
2 The authors forgot to put a minus in the Kissinger equation (2).
3 Thermal analysis researchers are well aware that the setup and specific details of the experiment will significantly impact the results. But this paper does not present the necessary information about dehydrogenation experiments under study. The required information is not included in [3,4]. The size and material of the crucible for loading the sample, its amount, the geometry and conductivity of reaction zone, the type of the furnace, the use (or absence) of additional thermal insulation, the location of the thermocouple, the characteristics of the thermocouple, and etc. are necessary. This is necessary for the understanding the results. Although the pre-hydrogenation procedure is described in [3], it is worth repeating. It is just a few lines.
4 Fig.1 is the same as in [3]. Can reference be made to previous research?
5 It is necessary to improve the style of the text and remove unnecessary repetitions.
6 Authors need to check the text for typos. For example, “6K/min” (Page 9, Line 209) should be replaced by “12K/min”.
7 There is a problem with terminology. Hydrogenation, dehydrogenation, dehydrogenation rate, hydrogenated sample and etc. are most common.
Author Response
This paper is devoted to the features of the dehydrogenation of home-made Mg-5Ni alloy in the Sieverts-type volumetric apparatus and the calculation of the kinetics of this process using the well-known Kissinger equation. The authors demonstrate a new empirical approach to estimation of Tm (temperature at which the reaction rate is highest) and to more accurate calculation of activation energy analyzing the function of dT/dt (T – temperature, t-time of reaction). I think that these results should be published in the Micromachines journal. The Sieverts-type reactors and the Kissinger equation are used quite often for different solid-phase dehydrogenation processes. These results are helpful for the readers. But major revision is necessary.
Major remarks
1 The paper contains only 8 references, 2 of them are the works of the authors. The content of the Introduction does not show the problem being solved by the authors. Some of this information is presented in the Discussion section at the end of the paper. The Kissinger kinetic model is the most used in studying the kinetics of the dehydrogenation of MgH2 and Mg-based alloys. What is the problem in such studies? Is this a scatter of the data in activation energy? Is this a big error in linear approximation? Is this an effect of the setup and experimental details on the results? Now, the information that the authors present in the Introduction is repeated several more times at the discussion of the results obtained. This part may be shortened as it contains generally understandable conclusions. I strongly advise the authors to rewrite the Introduction and expand the number of citations. Defining the requirements for the using the Kissinger equation and the already proposed ways to improve the calculation of kinetics using this model are useful.
Response: The Introduction was rewritten to show why we performed this work. References were added. The information that the authors present in the Introduction was shortened in the Discussion section.
There is no problem in applying the data of thermal analysis to the Kissinger equation to find activation energy for dehydrogenation. In this work, we proposed three methods to find activation energy for dehydrogenation by obtaining data in a Sieverts-type volumetric apparatus. This method has an advantage: data can be obtained in the same apparatus where samples were activated and hydrogenated.
The scatter of data for the plot to apply the Kissinger equation in our first and second methods is caused by errors from variation of room temperature, the thermocouple for the reactor, pressure transducers, and the distribution of temperature in the Sieverts-type apparatus.
In addition, we mentioned the requirements for using the Kissinger equation (i.e. In order that the Kissinger equation can be used to obtain the activation energy, the reaction rate equation must obey a first-order law.). Previously reported works did not show the obedience of the first-order law by the dehydrognation rate equation.
2 The authors forgot to put a minus in the Kissinger equation (2).
Response: It was corrected.
3 Thermal analysis researchers are well aware that the setup and specific details of the experiment will significantly impact the results. But this paper does not present the necessary information about dehydrogenation experiments under study. The required information is not included in [3,4]. The size and material of the crucible for loading the sample, its amount, the geometry and conductivity of reaction zone, the type of the furnace, the use (or absence) of additional thermal insulation, the location of the thermocouple, the characteristics of the thermocouple, and etc. are necessary. This is necessary for the understanding the results. Although the pre-hydrogenation procedure is described in [3], it is worth repeating. It is just a few lines.
Response: The following was added in the in the Materials and Methods section. The setup and specific details of the experiment were added:
The Sieverts-type volumetric apparatus consisted of a reactor part and a hydrogen supplying part having a known standard volume. Each part was connected to a pressure transducer. The temperature of the reactor was controlled by a programmed temperature controller. Between the reactor part and the hydrogen supplying part, a back-pressure regulator, which controlled the pressure of the reactor, was installed.
The reactor for loading the sample was in a shape of a cylinder made of S316 stainless steel with an outer diameter of 3/8’’ (9.5 mm) and a length of 130 mm. The amount of the used sample was 0.5 g. The conductivity of S316 stainless steel is 18.9 W/m K. The sample was heated in a vertical box furnace with block off for heat leak by 6T (3T+3T) silicon pad in entrance. The vertical box furnace was made by thermal insulation lightweight refractory bricks with ceramic fiber tube insulation in center position. A filter was installed in the upper part of the reactor to prevent sample loss or damage to the valve parts due to backflow of the sample in the vacuum process. Type K thermocouple probe of 1/8” diameter with stainless steel sheath was contacted along the cylinder wall of the reactor. The type K thermocouple can measure temperatures from 273 K to 1343 K.
The pre-hydrogenation procedure was described in the Materials and Methods section: Before dehydrogenation measurements, Mg-5Ni samples were hydrogenated at 593 K under 12 bar H2 for 60 min. The samples were cooled to room temperature in the furnace.
The hydrogen pressure (expressed in mV by transducer) in the hydrogen supplying part (to calculate the quantity of the hydrogen released under 1.0 bar H2) and the temperature of the reactor were measured as a function of time as the sample was heated at the heating rates (Φ) of 3, 6, 9, 12, and 15 K/min, respectively. The upper limit of temperature was set as 673 K. The variation of the hydrogen pressure was used to calculate Hd.
4 Fig.1 is the same as in [3]. Can reference be made to previous research?
Response: The reference of the previous research was given.
5 It is necessary to improve the style of the text and remove unnecessary repetitions.
Response: We tried to improve the style of the text and checked unnecessary repetitions
6 Authors need to check the text for typos. For example, “6K/min” (Page 9, Line 209) should be replaced by “12K/min”.
Response: The text was checked. 6K/min” (Page 9, Line 209) was changed to “12K/min”.
7 There is a problem with terminology. Hydrogenation, dehydrogenation, dehydrogenation rate, hydrogenated sample and etc. are most common.
Response: Hydrogenation, dehydrogenation, dehydrogenation rate, and hydrogenated sample were used instead of hydride formation, hydride decomposition, dehydriding rate, and hydrided sample, respectively.
Thank you very much for your detailed review, comments, and suggestions.

Round 2
Reviewer 1 Report
The authors have revised the manuscript well. I would like to suggest its publication in Micromachines in current version.
Reviewer 2 Report
The manuscript presents improvements compared to the 1st version; and can be accepted after minor (refinement) of the English spelling and grammar. Unfortunately, some experiments cannot be repeated. But the theoretical fundamentals are ok, and it is a good explanation of the results and leading causes of errors and deviations.
Reviewer 3 Report
The authors answered my questions and made the required changes. The paper may be published.